# An Overview of High Frequency Acoustic Sensors—QCMs, SAWs and FBARs—Chemical and Biochemical Applications [note 1]

**DOI:** 10.3390/s19204395

**Published:** 2019-10-11

**Authors:** Adnan Mujahid, Adeel Afzal, Franz L. Dickert

**Affiliations:** 1Department of Analytical Chemistry, University of Vienna, Währinger Straße 38, A-1090 Vienna, Austria; adnanmujahid.chem@pu.edu.pk (A.M.); aa@aafzal.com (A.A.); 2Institute of Chemistry, University of the Punjab, Quaid-i-Azam Campus, Lahore 54590, Pakistan; 3Department of Chemistry, College of Science, University of Hafr Al Batin, Hafr Al Batin 39524, Saudi Arabia

**Keywords:** quartz crystal microbalance (QCM), surface acoustic wave (SAW), film bulk acoustic wave resonator (FBAR), chemical sensors, biochemical sensors

## Abstract

Acoustic devices have found wide applications in chemical and biosensing fields owing to their high sensitivity, ruggedness, miniaturized design and integration ability with on-field electronic systems. One of the potential advantages of using these devices are their label-free detection mechanism since mass is the fundamental property of any target analyte which is monitored by these devices. Herein, we provide a concise overview of high frequency acoustic transducers such as quartz crystal microbalance (QCM), surface acoustic wave (SAW) and film bulk acoustic resonators (FBARs) to compare their working principles, resonance frequencies, selection of piezoelectric materials for their fabrication, temperature-frequency dependency and operation in the liquid phase. The selected sensor applications of these high frequency acoustic transducers are discussed primarily focusing on the two main sensing domains, i.e., biosensing for working in liquids and gas/vapor phase sensing. Furthermore, the sensor performance of high frequency acoustic transducers in selected cases is compared with well-established analytical tools such as liquid chromatography mass spectrometry (LC-MS), gas chromatographic (GC) analysis and enzyme-linked immunosorbent assay (ELISA) methods. Finally, a general comparison of these acoustic devices is conducted to discuss their strengths, limitations, and commercial adaptability thus, to select the most suitable transducer for a particular chemical/biochemical sensing domain.

## 1. Introduction

Chemical/biochemical sensors [1,2,3] are smart miniaturized devices having chemical or biologically derived recognition elements integrated with a suitable transducer that transforms the binding event information between sensor layer and analyte into a measurable electrical signal. The nature of the transducer plays a vital role in obtaining high sensitivity, faster response/recovery time, and low noise level. Moreover, the stability of the device is also critical against surrounding parameters such as viscosity, temperature, humidity and others. For on-field measurements [4], the size, design, data acquisition, and integration ability of transducer devices [5] are also considered necessary features. Apart from all these characteristics, one of the most important traits of an ideal sensor is the detection of target analyte without using any labeling indicator [6]. This means that target analyte should be analyzed as such based on its intrinsic or built-in features which may include optical, electrochemical, thermal, magnetic and other properties. Thus, for instance, optical transducers [7,8] would detect optical shifts resulted from analyte binding with the sensor interface. In the case where target analyte does not have any pronounced optical, electrochemical or other functionalities, it still can be recognized by acoustic devices [9,10,11,12,13]. Mass is the fundamental property of any analyte that can be monitored using acoustic or gravimetric devices which makes acoustic resonators as universal transducers.

Acoustic devices have been widely used for developing smart chemical and biochemical sensor applications. The principal advantage of using these transducers is their ability for label-free [14,15,16] recognition of target analyte without using any external reagent/chemical. This allows eliminating the labeling step thus, recognizing analyte exclusively based on to its intrinsic properties, thus reducing the cost and time of the labeling step. Modern sensor research is largely focused on label-free detection protocols and acoustic devices are highly suitable transducers to meet this requirement. Although there is a huge number of electrochemical [17,18,19] and optical sensing technologies [20,21,22] reported in literature for a variety of targets however, acoustic sensors [23] are well distinguished from the former types of devices due to their unique label-free detection feature, exceptionally high sensitivity i.e., down to pg level [24,25], miniaturized size as low as 1 mm or below and straightforward integration for wireless communication.

In 1959, Sauerbrey published his classical contribution [26] related to weighing thin films using quartz crystals, which established the basis of acoustic transducers for gravimetric sensing and other applications. The early stage use of acoustic devices was mainly for developing frequency filters, resonators, signal processing, actuating and others. The last two decades witnessed a significant increase in the use of acoustic wave devices [27,28,29] for chemical/biochemical sensing. These devices include mainly quartz crystal microbalance (QCM), film bulk acoustic resonators (FBARs), surface acoustic wave (SAW), shear horizontal surface acoustic wave (SH-SAW), shear horizontal acoustic plate mode (SH-APW) and shear transverse wave (STW) and flexural plate wave (FPW) devices. Anything that influences the wave propagation or cause surface perturbations at device interface, would lead to change the characteristic parameters of these devices including resonance frequency, acoustic wave velocity and other acoustoelectric properties. Based on wave propagation mode, acoustic wave devices are mainly categorized in two classes, i.e., bulk acoustic and surface acoustic wave devices. In bulk acoustic wave (BAW) resonators [30,31], the acoustic wave propagates through the piezoelectric crystal in thickness direction whereas, in SAW devices [32,33], the acoustic wave travels parallel to the surface of the piezoelectric substrate. Thus, QCM, FBARs, SH-APM are representative bulk acoustic devices while SAW, SH-SAW, and STW are surface acoustic devices.

There is a wide variety of receptor coatings that can be combined with acoustic devices for developing highly sensitive sensors. These receptors may range from natural antibodies [34,35,36,37], aptamers [38,39,40,41], DNA [42,43,44,45], protein/peptides [46,47,48,49] to synthetic affinity materials including functionalized polymeric layers [50,51,52,53], nanoparticles [54,55,56,57], carbon nanotubes [58,59,60,61], graphene oxide [62,63,64,65] and many others. As a result, the sensing domain of acoustic devices is significantly large which covers the detection of diverse bio-analytes such as bacteria [66,67], viruses [68,69] and whole cells [70,71,72,73], recognition of clinical biomarkers in complex samples [74,75,76], food analysis [77,78], process control and monitoring of environmental toxins [79] in liquid and gaseous phases. In terms of sensitivity and reliability, the sensor results of these devices are often comparable to classical analytical instruments such as liquid chromatography-mass spectrometry (LC-MS) systems, gas chromatography (GC), enzyme-linked immunosorbent assay (ELISA), polymerized chain reaction (PCR) and others. Moreover, the size of acoustic sensor devices is greatly reduced comparing to other large size instruments which enable them for on-field measurements. Unlike many established electrical sensors, the beneficial aspect of acoustic sensors is their ability to work under ambient environments which means that these devices do not require special conditions for example of temperature to operate thus, making detection of target analytes straightforward.

Among acoustic resonators, QCM [80,81,82] is one of the best known and most extensively studied gravimetric transducer. The highlight features of QCM devices are their low-cost, easy availability, simple design, and ability to work both in liquid and gaseous environments. The resonance frequency of QCM devices is usually in the range of 5–20 MHz [83,84]. Since QCMs are a type of thickness shear mode (TSM) resonators [85] and thus, higher frequency devices demand thinner quartz wafers, which eventually make QCM fragile and limit its applications. According to Sauerbrey [26], fundamental resonance frequency has prime importance in obtaining higher sensitivity and therefore, acoustic devices with high operating frequency are highly desirable. Surface acoustic devices such as SAW sensors [86,87,88] offer much high operating frequencies than QCM and consequently, result in enhanced sensitivity. However, they suffer from excessive damping loss in liquid media which makes them unsuitable for liquid phase sensing. FBARs [89,90] are typical BAW devices that have a similar structure and working principle like QCMs. However, their operating resonance frequency is even higher than SAW devices i.e., in GHz range. In this review article, we shall provide a concise overview of high frequency acoustic sensors going through QCM, SAW, and FBARs including their working principle, operating frequencies, selection of piezoelectric material and design fabrication, temperature-frequency relationship and working in liquid environments. The selected or showcased sensing applications of high frequency QCM, SAW and FBAR devices for biosensing as well as gas phase sensing would be exclusively covered. In selected cases, we shall compare the sensing performance of high frequency acoustic devices with established analytical techniques to demonstrate the potential of acoustic transducers. Each type of acoustic transducer has distinct features in respect to sensitivity, miniaturized design, ability to work in liquids, manufacturing costs, commercial adaptability, and fabrication issues. Therefore, apart from their significant advantages their main limitations are also discussed.

## 2. Acoustic Sensors QCM, SAW and FBARs

This section is primarily dedicated to the representative high frequency acoustic chemical sensors i.e., QCM, SAW and FBARs. Herein, we shall briefly describe their working principle, operating frequencies, selecting of piezoelectric materials for their fabrication, temperature-frequency dependence and operation in liquid phases.

### 2.1. Comparison of Working Principle

QCMs and FBARs belong to the family of BAW resonators while SAW devices are the typical surface acoustic resonators. QCMs are the first-generation acoustic resonators that are used for over two decades as gravimetric chemical/biochemical sensors. They are referred to as thickness shear mode (TSM) resonators that consist of thin AT-cut quartz wafer having symmetrically patterned circular electrodes on both sides. On applying a voltage between the electrodes, shear deformation takes place which makes QCM wafers highly sensitive to any surface perturbations [91]. Moreover, due to the propagation of shear waves in these devices, they do not radiate considerable energy thus, are suitable for liquid phase operations [92] without much damping loss.

Figure 1A shows an image of 195 MHz high fundamental frequency (HFF) QCM obtained from KVG Quartz Crystal Technology GmbH (Neckarbischofsheim, Germany). Figure 1B shows an image of 150 MHz HFF-QCM integrated with a customized printed circuit board (PCB) for improved handling of HFF-QCM device.

In SAW devices, acoustic wave propagation mode can be described by Rayleigh waves as explained by Lord Rayleigh [95]. These waves have two components i.e., longitudinal and shear vertical component which can couple with the layer material in contact at device interface. The coupling between layer material and the device strongly influences acoustic wave propagation. This means that acoustic energy is highly confined at device interface and therefore any surface interactions with layer material make these devices extremely sensitive to mass loadings, conductive changes and other factors. The penetration depth of these waves is in the range of a few acoustic wavelengths and does not depend on the entire thickness of the substrate. In a typical SAW design, interdigital transducer (IDT) electrodes [96,97] are fabricated on piezoelectric material where on the application of electric voltage, acoustic waves are generated that travel across the substrate surface. By covering these IDTs with suitable recognition layer, these devices can be used for chemical sensing. Figure 2A shows a high-resolution image of a SAW device captured by an optical microscope, while Figure 2B displays a magnified image showing the geometrical details of IDTs. Figure 2C shows images of different SAW resonators having frequencies 100, 200, 433 and 1000 MHz. As the frequency of the device increases from 100 to 1000 MHz, the size decreases from 8 to 1 mm. This suggests that an increase in resonance frequency demands a more compact design. The dimensions of IDTs including length, width, height and spacing between IDTs fingers can be optimized to enhance the sensing performance of SAW devices. SAW devices are well known for gaseous phase sensing and found a number of applications including commercial sensors [98,99] as well. However, when these devices are immersed in the liquid phase, compressional waves are generated which suffers from high attenuation loss of acoustic energy. This issue is addressed by adjusting the cutting angle of the piezoelectric material so that vertical component of acoustic waves is changed to shear horizontal polarized surface acoustic waves (SH-SAW) resulting in a significant decrease in device’s attenuation when used in liquids [100,101]. SH-SAW devices are also considered as shear transverse wave (STW) resonators which also found a number of liquid phase sensing applications [102,103].

The basic working principle of FBARs devices is similar to that of QCMs however; these devices offer much higher resonance frequency with smaller base mass than QCMs which makes them superior in terms of sensitivity. Furthermore, their miniaturized design and simple operation are highly suitable for integrating with electronic systems. The fabrication design, nature of substrate and transduction mechanism of FBARs is different from QCMs. As in case of FBARs a thin film of the piezoelectric material is sandwiched between two metal electrodes and on applying an electric voltage across the electrodes, a standing wave is generated in piezoelectric layer resonating at a frequency depending on the thickness of the piezoelectric layer. The piezoelectrically active area is isolated from substrate material so that acoustic waves generated may not radiate into the substrate thus, losing any resonance. Based on isolation methods, there are two structures of FBAR: in the first case an air cavity is designed for acoustic isolation of piezoelectric area from the substrate while in the second type acoustic Bragg reflectors are designed i.e., consisting of alternating layers of low and high acoustic impedance. The latter type of devices is also called solidly mounted resonators (SMRs) [105]. Figure 3A shows a schematic diagram of a FBAR device while Figure 3B represents SMR design. Figure 3C shows packaged FBAR device connected with a network analyzer and integrated with customized PCB. There are two resonating modes [106] in FBARs devices i.e., thickness longitudinal mode (TLM) and thickness shear mode (TSM). In the first type, longitudinal waves are generated while in second type shear waves are produced. The basic difference between these modes is the orientation of piezoelectric crystal, i.e., *c*-axis angle [107,108], as in the case of TLM where the *c*-axis angle of crystal orientation is perpendicular to the substrate material. Whereas in TSM, the crystal is planar, i.e., inclined around 34.5°, which produces shear waves. In TLM, the acoustic waves suffer from high attenuation loss when used in liquid media and thus, mainly used for gaseous phase sensing, while TSM devices exhibit low damping and thus, better suited for liquid phase operations.

### 2.2. Comparison of Operating Frequencies

Typical resonance frequency of QCMs is in the range of 5–20 MHz [110,111] with most of the devices having 10 MHz. Since QCMs are examples of TSM resonators [112], their resonating frequency depends on the thickness of piezoelectric substrate. Therefore, reducing thickness beyond a certain point would make them mechanically unstable and fragile. This is a major limitation in producing QCMs of high fundamental resonance frequencies. This problem has been addressed by developing the central part around the circular electrode area of QCM thinner while the outer frame is kept thick which is mechanically stable. The strategy is named as “inverted mesa” or “biconvex” design [113] and such devices are called HFF-QCMs [114]. This technique leads to the fabrication of ultrasensitive QCMs sensors having fundamental resonance frequency as high as 200 MHz. These devices can be manufactured by different technologies including wet etching, deep reactive ion etching (DRIE), i.e., gas phase etching and metal sputtering methods. The wet etching is relatively an inexpensive method whereas DRIE is more expensive but offers highly anisotropic etching. Q-factor or quality factor is a dimensionless parameter that describes the damping spectra of an acoustic resonator. The decrease in the thickness of quartz wafer increases the Q-factor which can be explained on the basis that by decreasing the thickness, acoustic energy trapping [115] is improved. Moreover, thinner QCMs offer good electrical conductivity and reduce the unwanted or spurious vibrational modes.

The fundamental resonance frequency of first generation SAW devices was initially in the range of few hundreds of MHz and nowadays with the ensuing technological advancements, the resonance frequency has been increased substantially up to the GHz range. In SAW devices, the fundamental frequency depends on IDT electrode geometries including the spacing between IDT fingers. Seidel and Hesjedal [116,117] explained that in order to achieve 5 GHz frequency of SAW device, the electrode geometries of around 0.15 µm dimensions (lines and the gaps) are needed to be developed while having two electrodes per wavelength with equal widths and gaps and an acoustic wave velocity of about 3000 m/s. The electrode geometries of 0.15 µm are difficult to achieve by conventional photolithographic processes since these methods are limited to produce electrode dimensions of around 0.5 µm. Therefore, in order to develop smaller electrode dimensions for developing high frequency SAW devices, alternative fabrication techniques such as nanoimprint lithography [118], electron beam lithography [119] and others could be adopted. Due to the restricted IDTs geometries, they are operated at higher frequency harmonics i.e., generated by low fundamental frequency sources. Nevertheless, in conventional device structures, the efficiency of SAW excitation decreases at higher harmonics thus, limiting their operation at higher GHz. In addition to IDT’s geometries, the nature of piezoelectric substrate is also decisive for enhanced acoustic wave velocities that can lead to higher operating frequencies.

Although QCMs and FBARs, both are BAW devices and have same working mechanism, the operating resonance frequency of FBAR devices is usually in the range of sub GHz to around 10 GHz [120], which is substantially higher than QCMs. Moreover, FBARs also have higher frequencies than SAW as the former device offers higher GHz frequency range compared to a few GHz offered by SAW. There are two important factors that contribute towards the high resonance frequency of FBARs i.e., the thickness and the nature of piezoelectric material. The thickness of piezoelectric films is in the range of sub-µm to few µm e.g., 0.5–3 µm [121] as it ultimately leads to a small base mass of piezoelectric substrate which yields high resonance frequency. The high sensitivity of FBARs is mainly driven by the fundamental resonance frequency and Q-factor which characterizes the bandwidth of resonance phenomena. The small frequency shifts are more accurately determined by FBARs having high Q-factor with sharp resonances than FBARs with low Q-factor thus, ultimately leading to enhanced sensitivity. Moreover, high Q-value is an indication of adequate acoustic energy trapping in the piezoelectric layer.

### 2.3. Selection of Piezoelectric Material

There is a wide range of piezoelectric substrates both natural and synthetic used for acoustic devices, nevertheless, quartz is the fundamental piezoelectric material that is most widely and frequently used for acoustic sensors, especially for QCMs. The selection of typical piezoelectric material and a specific cutting angle is vital in developing high frequency acoustic devices. The use of quartz as first choice piezoelectric material is because of its high Q-value, simple design, easy availability and low fabrication cost. AT-cut quartz i.e., singly rotated Y-cut having its normal axis parallel to *y*-axis with θ ≅ 35.25° [122] is the most common design of QCMs for sensor applications.

Apart from quartz, lithium tantalate (LiTaO_3_) and lithium niobate (LiNbO_3_) are the most common piezoelectric materials used for SAW manufacturing. LiTaO_3_ (36° Y-X) and LiNbO_3_ (64° Y-X) offer higher wave velocities than quartz and are therefore suitable for manufacturing high frequency SAW devices. Aluminum nitride (AlN) is a suitable piezoelectric material [123] that shows high acoustic wave velocity and are suitable for developing high frequency SAW devices. The AlN/diamond films [124] are known for remarkably high acoustic velocity i.e., around 11,000 m/s which offers high electromechanical coupling constants that leads to the development of high frequency SAW devices. Kirsch et al. [125] used AlN/diamond layered structures for SAW fabrication having a frequency of 5 GHz. The width of IDT electrodes was 0.5 µm developed by electron beam lithography.

Like other acoustic devices, the selection of piezoelectric material for fabrication of FBAR is an imperative step to have both high resonance frequency and improved Q-value. Zinc oxide (ZnO) [126] and AlN [127] thin films are the two most extensively studied piezoelectric materials used in the development of FABRs due to their enhanced electro-acoustic features. The structural properties of piezoelectric material such as crystallographic orientation, surface morphology and electrical impedance have a significant influence on FBARs performance. As mentioned above, AlN offers high acoustic wave velocity that yields high frequency FBAR devices. In addition to this, AlN interface is inert and biocompatible thus, suitable for integrating with biological receptor materials for biosensing applications. Garcia-Gancedo et al. [128] showed the growth of carbon nanotubes (CNTs) on AlN surface to use them as top electrodes and compared its performance with metal i.e., Au electrodes. The resulted device had shown higher Q-values indicating that CNT electrodes had a lower acoustic loss and higher acoustic energy trapping at the piezoelectric layer. Moreover, the surface of CNTs can be functionalized with chemical or biological derived layer for enhanced sensitivity and selectivity in typical sensor applications. In comparison to AlN, ZnO thin films possess finer and smoother surface, high electromechanical coupling coefficient and relatively easy to synthesize. On the other hand, the acoustic velocity in ZnO is comparatively less and it is not compatible for fabricating with microelectronic devices such as a complementary metal oxide semiconductor (CMOS) which somewhat limits its applications. Lead zirconium titanate (PZT) [129] is another choice of piezoelectric material with high electromechanical coupling coefficient than AlN and ZnO. However, on the other hand, PZT shows higher acoustic attenuation, lower acoustic velocity, and inadequate biocompatibility which limit its use in sensor applications. In general, AlN and ZnO are the two most common piezoelectric materials used for FBAR fabrication. Nevertheless, the choice between them mainly relies on fabrication conditions and ultimate applications of device.

### 2.4. Temperature-Frequency Dependency

The acoustic wave velocity is temperature dependent and the nature of piezoelectric material, crystal orientation and cutting angle are important factors that influence frequency shifts due to change in temperature. While selecting a typical piezoelectric material for developing high frequency acoustic devices, it is equally important to consider temperature-frequency relationships [23]. The frequency shifts due to temperature fluctuations should be as minimal as possible during sensor measurements. Quartz is advantageous in this regard, as it shows negligible or almost zero temperature-frequency shifts around room temperature. The maximum working temperature for quartz is much high [130] suggesting that quartz-based acoustic devices are stable in high thermal stress. AT-cut quartz crystals i.e., the most popular material for QCMs remains stable against temperature shifts as long as they are bare. However, when certain coating material is applied to these devices, they no longer remain insensitive to temperature. Moreover, temperature sensitivity becomes more serious when a chemical/biological layer coated QCM is exposed to liquid environments. The suitable way to address this issue is introducing a reference electrode/channel to compensate frequency shifts due to temperature fluctuations.

ST-cut i.e., stable temperature cut quartz has nearly zero temperature coefficient of frequency (TCF) for developing SAW sensors, which means that such SAW devices are nearly stable against temperature shifts. LiTaO_3_ and LiNbO_3_ are the two most frequently used piezoelectric materials for SAW fabrication. These materials show linear temperature-frequency relationship which indicate that the increase in temperature would increase in frequency shifts. The crystal orientation and the cutting angle of these piezoelectric materials also have influence on TCF values. SAW devices as physical sensors are widely used for monitoring temperature shifts [131], however, the use of SAW for chemical or biochemical sensing require temperature compensation element to figure out temperature frequency shifts.

In the case of FBARs, AlN and ZnO are sensitive to temperature shifts, having TCF values [132] of 25 and 40 ppm/°C, respectively. In order to compensate for frequency shifts due to temperature, the use of an additional compensation layer of a certain piezoelectric material is reported in literature [133]. The main drawback of this strategy is that the overall thickness of the piezoelectric layer increases which ultimately decreases the fundamental resonance frequency as well as the electromechanical coupling coefficient. Bjurstrom et al. [134] reported that the temperature compensation of second harmonic shear mode in a composite thin film of Al/AlN/Al/SiO_2_ resulted in high Q-value and high fundamental resonance frequency. Apart from using composite arranged layers, Pang et al. [135] reported the fabrication of micro-designed air-gap capacitor which passively minimizes TCF about 40 ppm/°C. When the temperature increases, the airgap of the capacitor also increases due to a difference in thermal expansion coefficients of the bimetallic layer which eventually decreases the capacitance. The authors reported that nearly zero TCF could be achieved by this strategy using AlN derived FBARs.

### 2.5. Operation in Liquid Phase

All the acoustic devices work efficiently as sensors in gaseous conditions. However, for liquid phase measurements, the liquid properties such as density, viscosity, conductivity and surface charge strongly influence the oscillations of acoustic resonators and thus, results the damping of acoustic energy. QCMs are TSM resonators that generate shear acoustic waves that do not radiate a considerable amount of energy in the liquid phase. Thus, these devices exhibit low damping and suitable resonance. QCMs are the most extensively studied acoustic sensors for liquid phase operations [136,137] and reported for diverse applications. Kanazawa and Gordon [138] studied the influence of liquid properties i.e., viscosity and density on QCM oscillation and derived a mathematical relationship for calculating the shift in fundamental resonance frequency. Later, Martin et al. [139] characterized QCM behavior under simultaneous liquid and mass loadings. The fundamental resonance frequency was decreased by both in contact liquid and surface mass and it was important to differentiate frequency shifts due to these factors. The authors had shown that with the help of an electromechanical model, the change in surface mass can be distinguished from changes in solution properties. Recently, Kang et al. [140] reported an associated high frequency resonance (HFR) model to study QCM operations in the liquid phase. The HFR model explains that the total resonance is resulted from solvent capacitance and inductance, static capacitance of quartz wafer and inductance of leading wires. The authors monitored a typical cell adhesion process and discussed the responses from HFR and TSM peaks. They figured out that the combined application of HFR and TSM is a suitable way to enhance the stability of QCM signals in liquid phase and thus, obtaining reliable sensor data.

In case of SAW devices, the liquid phase operation is seriously hampered due to the coupling of the longitudinal component of Rayleigh waves with the in-contact liquid. This coupling leads to high attenuation in amplitude and velocity of surface waves. In addition to this, using quartz as the piezoelectric material for SAW, causes serious problems in aqueous phase due to the considerable difference in dielectric constants of quartz (3.8) and of water (78). This mismatch in dielectric constants leads to no resonance signal when high frequency is applied. Keeping in view the above issues, there is a need to explore alternative piezoelectric materials with high dielectric constant and suitable cutting angles as well to study other wave propagation modes that don’t suffer from acoustic energy loss in the liquid phase. Shiokawa and Moriizumi [141] studied SAW operation in liquids using the leaky SAW mode that has shear horizontal propagation. Their findings revealed that cutting angle of piezoelectric material, i.e., LiTaO_3_ has to be rotated around 36° so that acoustic wave propagation mode changes form vertical to shear-horizontal mode that considerably reduces the acoustic loss. Furthermore, LiTaO_3_ (36° Y-X) having a sufficiently high dielectric constant of about 43 which is suitable piezoelectric material for liquid phase sensing. LiNbO_3_ is another choice of piezoelectric material for fabricating SH-SAW devices that has much higher electromechanical coupling constant. However, the acoustic loss is relatively higher. With the proper selection of the piezoelectric substrate and crystal cutting angle, SH-SAW also called as STW device has been reported for liquid phase sensing applications. Love wave acoustic devices are a typical example of SH-SAW which exhibit high sensitivity and low attenuation losses when exposed to liquid phase [142]. In Love wave sensors [143,144], a special wave guiding layer is coated so that acoustic energy is confined in this layer while a chemical recognition layer is applied on top of this guiding layer. The thickness of the guiding layer is important to achieve the optimal performance.

FBAR devices as mentioned above have two main resonating modes i.e., TLM and TSM with only difference in piezoelectric crystal orientation. In TLM devices, longitudinal waves propagate which suffer from severe damping in liquids leading to a significant reduction in Q-values and thus, limit its operation to dry environments mainly. Whereas TSM devices exhibit strong shear resonance in liquids with little damping resulting in high Q-value and thus, is suitable for liquid phase sensing. There are a number of liquid phase sensor applications of shear mode FBAR devices including biosensing and others. Zhang et al. [145] reported that for FBARs in liquids, the Q-value at second harmonic resonance is significantly higher than at fundamental resonance frequency. In a later report [146], the authors investigated the impact of liquid properties such as density and conductivity on the resonance frequency of FBAR devices. The authors showed that with increasing density of solvents i.e., acetone, water and dimethyl sulfoxide; both the parallel and series resonance frequencies decreased. This can be explained in a way that density and mass have a direct relationship thereby, causing a decrease in frequency at increasing liquid density. While the increase in conductivity of loaded liquid demonstrated a decrease in series resonance frequency but an increase in the parallel frequency. The authors proposed that this converse behavior of resonance frequencies due to the conductivity of liquid could be used to distinguish the resonance frequency shifts due to mass loading from the conductive effects.

Table 1 summarizes an overview of high frequency QCM, SAW and FBAR devices describing the nature of acoustic wave type, operating frequency, sensitivity, selection of piezoelectric materials and their characteristics, working medium, availability, manufacturing cost, commercial adaptability and some general comments about these devices.

## 3. Exemplary Sensor Applications

In this section, the selected sensor applications of high frequency QCMs, SAW and FBARs in diverse fields would be briefly reviewed. The two main sensing domains i.e., biosensing related to liquid phase sensor applications and the gas phase sensing would be exclusively covered. Furthermore, we shall compare sensor characteristics of high frequency acoustic devices with other analytical strategies where appropriate.

### 3.1. Biosensors

A typical biosensor is a device which has a biochemical recognition layer integrated with a transducer that converts the binding event information into a measurable signal. Here, we shall discuss high frequency acoustic transducers i.e., QCM, SAW and FBAR for biosensing applications. Uttenthaler and coworkers [149] used 39, 56, 70 and 110 MHz HFF-QCMs and compared their results with 19 MHz QCM for the detection of bacteriophage M13. The 70 and 110 MHz HFF-QCMs showed much higher noise while 56 MHz device exhibited the best signal to noise ratio. The authors reported that by using 56 MHz HFF-QCM, the detection limit was enhanced by a factor of 200 as compared to 19 MHz device, which is remarkable in view of improving sensitivity and lowering detection limits in immunoassays. Fernandez et al. [94] used 150 MHz HFF-QCM for studying binding interactions between protein A and IgG antibodies. The authors carried out finite element method (FEM) simulations to improve the geometrical fabrication design of the developed sensor thus, to have optimized electrical response. The numerical data from FEM simulations were in good correlation with the experimental characterization of acoustic sensor. Using 150 MHz HFF-QCM, the authors claimed a combined frequency shift of 51,000 Hz for protein A adsorption and IgG injection while in literature the reported frequency shifts for a 9 MHz QCM sensor was only 200 Hz for the similar studies. March et al. [150] developed HFF-QCM immunosensor having frequency 100 MHz for the detection of pesticides i.e., carbaryl and thiabendazole. The authors compared the analytical performance i.e., assay sensitivity (*I*_50_ value), limit of detection (LOD) and working range (WR) of HFF-QCM immunosensor with conventional low frequency QCMs, SPR and ELISA methods. The comparative data of standard and optimized ELISA, SPR and QCM sensors for carbaryl detection is shown in Table 2. Here, the higher *I*_50_ value indicates lower sensitivity. For carbaryl sensing, they reported that 100 MHz HFF-QCM immunosensor exhibited substantially higher sensitivity, lower detection limit and wider detection ranges than SPR and QCMs with fundamental frequencies 9, 10 and 50 MHz. While comparing the data with ELISA, the HFF-QCM sensor performance is somewhat less than optimized ELISA but still closely related to the results of standard ELISA method. In a similar way, 100 MHz HFF-QCM sensor for thiabendazole exhibited better detection limit than SPR.

In a later report [151], they extended their previous strategy for the detection of carbaryl residues in honey samples and compared the HFF-QCM sensor results with liquid chromatography tandem mass spectrometry (LC-MS). Figure 4 shows the comparison of HFF-QCM and LC-MS data for the detection of carbaryl in fortified honey samples. The analysis of spiked honey samples by HFF-QCM showed comparable detection limits and quantification to LC-MS for carbaryl detection. Moreover, the developed sensor requires minimal sample pre-treatment protocols and thus, suitable for rapid analysis of carbaryl. These findings indicate that HFF-QCM sensors could be potential alternatives to classical instruments such as LC-MS. In another effort [152], the authors combined 100 MHz HFF-QCM with specified monoclonal antibodies for highly sensitive detection of tuberculosis biomarker i.e., 38 kDa protein from *Mycobacterium tuberculosis*. The analytical data of HFF-QCM sensors regarding sensitivity, limit of detection and working ranges are better than previously reported ELISA results. Furthermore, the detection range of developed immunosensor is comparable to tuberculosis antigens found in patient samples suffering from this disease.

Länge et al. [153] comprehensively reviewed the biosensing applications of SAW resonators including STW and SH-SAW devices. The authors explained that suitable surface modification of SAW transducers is highly important for successful biosensor applications. They described different protocols for coating the recognition layers at SAW interface such as using direct immobilization of analyte-specific layers, immobilization of antibodies via protein A/protein G, functionalized dextran or hydrogel layers and others. The reported sensor applications cover the detection of antibodies, bacteria, DNA and various small molecules including polycyclic aromatic hydrocarbons, atrazine and others. Most of the SAW biosensors reported in literature had fundamental resonance frequency less than 1 GHz. There are relatively few articles reporting GHz-frequency SAW devices for biosensing applications. For instance, Krishnamoorthy et al. [154] functionalized monoclonal interleukin-6 (IL-6) antibodies on to 0.7 GHz and 1.5 GHz SAW devices for the detection of IL-6. Since IL-6 is an important protein that helps in regulating immune responses and its elevated levels are associated with various diseases including cardiovascular disease, diabetes, arthritis and certain types of cancer therefore, its detection in blood has high importance. The authors studied three different approaches to integrate IL-6 proteins at SAW surface i.e., (a) by direct surface adsorption, (b) surface modification through bovine serum albumin (BSA), and (c) surface modification via monoclonal antibodies. It was noticed that direct surface adsorption is not an effective approach for adequate protein recognition and thus, resulted in low sensitivity for IL-6 detection. The second method was IL-6 immobilization through BSA which showed much higher frequency shifts for IL-6 detection than simple surface adsorption. While the third strategy i.e., IL-6 immobilization through monoclonal antibodies exhibited highest sensor shifts for IL-6 detection. Figure 5 showed the sensor responses of two 1.5 GHz SAW devices coated by IL-6 immobilization through BSA (Figure 5A) and IL-6 immobilization through monoclonal antibodies (Figure 5B). It can be seen that IL-6 immobilization through antibodies exhibited higher sensitivity compared to immobilization through BSA. Moreover, from these results it can also be noticed that by increasing the active sensor area i.e., 20 × 20 µm^2^, the detection range can be improved comparing to 5 × 5 µm^2^. Finally, the authors tested the developed SAW biosensor for the detection of IL-6 in human serum and found that it showed satisfactory results in agreement with ELISA for detecting low concentrations of IL-6.

Kim et al. [155] reported 2.4 GHz SAW biosensor having biotin–streptavidin and DNA hybridized biomolecular binding interface. The authors reported low detection limit i.e., ~ 1 ng/mL and rapid response for the detection of target analyte with the possibility of wireless sensor integration. Cai et al. [156] developed a highly sensitive SAW biosensor utilizing the third-order harmonic mode of 2 GHz device to achieve resonance frequency of 6.4 GHz with Q-factor of more than 4000. They reported appreciable sensitivity for target DNA detection i.e., 6.7 × 10^−16^ g/cm^2^/Hz with the ability of single cell detection which is remarkable. Furthermore, a linear sensor curve was recorded in the concentration range from 1 µg/mL to 1 ng/mL. The developed biosensor exhibited high selectivity as shown in Figure 6 where it can be seen that the response of target DNA (Sample 1) having concentration 1 µg/mL with 15 bases has highest relative frequency shift. Sample 2 has the same concentration with 15 bases but only has one mismatch base thus, showed lower frequency shift than 1. In the same way, Sample 3 had the same concentration with 15 bases but 2 mismatch bases which result in even lower response. Samples 4 and 5 having concentrations of 1 µg/mL each represent the responses of 4 and 3 bases, respectively. Samples 6, 7, and 8 show the sensor responses of target DNA with 15 bases having concentrations 100, 10 and 1 ng/mL, respectively. Finally, Sample 9 shows the response of BSA i.e., an interfering analyte and the Sample 10 represents PBS buffer response. The proposed sensor setup has the potential for label-free detection of target analytes with high sensitivity which can be used for medical diagnostics and other applications.

Weber et al. [12] explained the operation of FBAR in liquid phase specifically for biosensor applications. They experimentally showed that sensor performance of shear-FBAR is much better than longitudinal-FBAR device due to lower Q-value and high noise of later type of device. The mass resolution or the smallest detectable mass by shear-FBAR device was better than QCM setup under the same conditions.

A recent review related to biosensing applications of FBARs has been published by Zhang et al. [157] in which the authors described theoretical aspects of FBAR’s working and operations in different environments and furthermore, proposed some optimization in the design. Additionally, the authors exclusively discussed biosensing domain of FABRs for various bio-analytes using immobilized antibodies, aptamers, modified DNA probes, enzymes and others as receptor materials. Like SAW devices, the authors explained that the surface density of immobilized antibodies and their orientation greatly influence the target antigen binding and ultimately affecting sensor performance. Gabl et al. [158] integrated DNA oligomers (25 bases) via thiol-modification at gold electrode interface of 2 GHz FBAR device. The authors used this setup for detecting protein as well as DNA hybridization. They reported a high mass sensitivity i.e., 2400 Hz cm^2^/ng which was calculated about 2500 times higher than a 20 MHz QCM. Lin et al. [159] monitored DNA synthesis using a 3.16 GHz FBAR device coated with self-assembled thiol-modified DNA primer. The authors reported that the developed setup is capable of monitoring the addition of single nucleotide which can be used for advanced DNA sequencing applications. In a later study [160], the authors reported the detection of cancer biomarker i.e., prostate specific antigen (PSA) using immunoglobulin G (IgG) antibodies as receptors. The orientation of IgG antibodies at the gold electrode was improved by immobilizing antibodies via protein G with suitable cross-linker. The resulted sensor exhibited high sensitivity i.e., 25 ng/cm^2^ using a 3.6 GHz device and exceptional low detection limit in the range of ng/mL. Moreover, the sensor showed suitable selectivity when exposed against BSA as even 100 times higher concentration of BSA than PSA, did not show appreciable frequency shifts. After rebinding, the antibody-antigen complex at sensor interface can be removed easily by suitable washing with glycine and buffer. The resulted sensor surface can be reused for further measurements as the regenerated surface showed comparable frequency shifts to that of fresh sensor coating. In another report, Zhao et al. [161] reported the detection of human prostate specific antigen (hPSA) using a 1.5 GHz FBAR device with even better sensitivity i.e., 1.5 ng/cm^2^ and higher Q-factor i.e., around 800. The authors utilized mouse monoclonal anti-prostate specific antigen (anti-hPSA) as receptors with an optimal amount of 1 mg/m^2^. They suggested that an increased antibody amount than optimal resulted in increased steric hindrance causing reduced binding efficiency while the lower antibody concentration led to a limited number of binding sites. In Figure 7, the authors showed the adsorption of h-PSA antibodies on FBAR electrode (1st column) resulting frequency shift of 230 kHz, then following BSA blocking leads to further 70 kHz shift (2nd column) and finally, h-PSA antigen binding resulted in 46 kHz frequency shift (3rd column). In control experiments, BSA adsorption of FBAR electrode resulted in 71 kHz (4th column) which is consistent with the frequency shift of blocking step i.e., 70 kHz as shown in the 2nd column. Further h-PSA antigen binding on top of BSA resulted only 2 kHz (5th column) suggesting the non-specific interactions of antigen with BSA. These findings indicate the potential of FBAR as the label-free sensor that can be used for highly sensitive and selective detection of antibody adsorption and it’s binding with antigens. Chen et al. [162] reported the detection of alpha-fetoprotein (AFP) i.e., a cancer biomarker using 2.1 GHz FBAR. The authors used monoclonal anti-AFP antibodies and reported a detection limit of 1 ng/mL of AFP antigens which is comparable to ELISA. The developed sensor showed suitable selectivity when exposed to other endogenous substances i.e., alcohol, ascorbic acid, uric acid and glucose.

### 3.2. Gas Phase Sensing

Chemical sensing of gaseous analytes by acoustic transducers [163] has been widely recognized due to high sensitivity, lower detection limits and small size for miniaturization and portability issues. The enhanced sensitivity is primarily driven by the high fundamental resonance frequency of the device. However, high frequency also leads to high damping which requires improved oscillation stability. The main advantage of using high frequency devices for gas phase sensing is their relatively low damping than liquids and thus, having better Q-factor. Fanget et al. [164] reviewed different types of electromechanical transducers for gravimetric detection of gases/vapors. They explained the operating principle, gas adsorption and detection mechanism, and important design details for improved gas sensing.

Waldvogel and coworkers [93] designed a prototype mobile handheld sensor setup having 195 MHz HFF-QCM for detecting airborne analytes. In order to improve the signal processing of HFF-QCM and its suitability for real-time applications, the authors investigated the performance of the device under different conditions including pressure and turbulence, static and dynamic temperature influence and acceleration-sensitivity. The authors aimed to improve the signal to noise ratio in continuous airflow and to minimize the impact of environmental conditions so that HFF-QCM setup could be used as a portable hand-held device. In another report [165], they developed 200 MHz QCM device coated with modified dendrimeric structures for online detection of triacetone triperoxide (TATP) i.e., an explosive. With such a high resonance frequency, a detection limit of 1 ppm for TATP was achieved with suitable response/recovery times i.e., <5 s. In order to enhance selectivity by reducing the non-specific interactions, the authors [166] modified HFF-QCM electrode surface with alkylated phosphonic acids and corresponding partially fluorinated phosphonic acid groups. The interesting aspect of these modifications is their hydrophobic and lipophobic properties as it resulted in a 55% reduced frequency shifts for water and 74% for cyclohexane, respectively. They reported that QCM surface modification reduces the nanoporosity and polarity of alumina (electrode material) and furthermore, allows suitable integration of supramolecular affinity layers on modified QCM surface for TATP detection. Principal component analysis (PCA) of the developed sensor array for TATP detection has demonstrated that after surface modification the sensor signals for TATP are much better separated from common interferents such as water, acetone, hydrogen peroxide and tBuOOtBu i.e., an analogous compound to TATP. Figure 8A,B demonstrate first principal component (PC1) and second principal component (PC2) data of un-modified and surface modified HFF-QCM sensor array for TATP detection, respectively.

SAW devices have been extensively utilized for gas phase sensing more than any other acoustic transducers. This is simply due to the inherent suitability of Rayleigh waves in gaseous environments, high resonance frequency that leads to low detection limits, fast response time and ability to integrate with miniaturized electronic systems. These features make SAW sensors highly competitive for developing viable gas sensors for commercial applications. Since the early work of Wohltjen [167,168], SAW devices as gas sensors have been widely investigated. Afzal et al. [169] reviewed a variety of different molecular recognition materials that can be combined with SAW transducers for developing fast and highly sensitive gas/vapor sensors. These include metal oxide nanostructures, carbon nanotubes (CNTs), graphene oxide hybrids and composites, molecularly imprinted polymers, supramolecular structures, polymeric affinity layers, self-assembled monolayers and others. The principal selection criteria for SAW sensor coating is its selectivity, response/recovery time, low cross-sensitivity against humidity and temperature shifts and ability to work under ambient conditions. Dickert and coworkers [170] showed that 1 GHz SAW device coated with permethylated β-cyclodextrin structure attached with hexafluorobenzene could detect *m*-xylene down to 1 ppm. Furthermore, they demonstrated that increasing the resonance frequency of SAW from 80 MHz to 1000 MHz, the sensor response increases in a parabolic way whereas the noise increases linearly. Figure 9A shows the sensor responses by 80, 301, 433 and 1000 MHz devices having the same coatings. A similar trend is also reported by Abraham et al. [171] as they integrated hybrid coatings of ZnO-CuO and CNTs for the detection of different volatile organic compounds (VOCs). The designed ZnO-CuO-CNTs nanocomposite exhibited enhanced sensitivity than pristine CNTs. The authors compared the sensor responses of different SAW devices having frequencies 80, 315, 433, 915 and 1000 MHz as shown in Figure 9B. It can be seen here that an increase in resonance frequency leads to increase in the sensor response which is in agreement with previous studies [170].

Liu et al. [172] combined inorganic and organic materials i.e., CeO_2_ nanoparticles with polyvinyl pyrrolidone nanofibers, respectively by electrospinning method and integrated as SAW sensor coatings. The authors used a 1.56 GHz SAW device and tested for relative humidity sensing in the range of 11 to 95%. They demonstrated a frequency shift of 2.5 MHz using 1.56 GHz device which was about 8 times higher than 0.879 GHz SAW having the same sensor coatings thus, showing higher resonance frequency leads to enhanced sensor shifts. Moreover, the designed sensor showed fast response/recovery kinetics, suitable repeatability of sensor signals and good stability of coating material even after one month. In a parallel report [173], the authors tested Ni(SO_4_)_0.3_(OH)_1.4_ i.e., (NSOH) nanobelts and NiO nanoparticles coated with 1.54 GHz SAW devices for relative humidity measurements, respectively. They compared relative humidity sensing data of developed sensors with other strategies reported in the literature, as shown in Table 3.

This table again confirms the frequency-sensitivity relationship as increased resonance frequency leads to enhanced sensor response. Additionally, the nature of coating material also plays important role in improving sensitivity as the response of NiO nanoparticle (5.81 MHz) coated SAW against relative humidity is higher than the response of NSOH nanobelts (2.95 MHz) coated device.

Due to the higher frequency and compact size, FBARs have turned out to be highly suitable for gas sensing applications as FBAR gas sensors offered much lower detection limits than other gravimetric sensors. Zhang and Kim [178] discussed the operation of FBAR mass sensor in liquid as well as in gaseous phases. They developed FBAR on silicon nitride diaphragm having ZnO piezoelectric film which was sandwiched between two aluminum electrodes. The measured mass sensitivity of FBAR was 726 cm^2^/g i.e., close to the theoretical value of 773 cm^2^/g and about 50 times higher than 6 MHz QCM. FBARs devices of 1 GHz showed higher Q-value i.e., 200–300 in air and can detect a mass change of 10^−9^ g/cm^2^. Whereas FBARs in liquids having frequency 2 GHz, showed Q-value 40 at 2nd harmonic resonance and can detect a mass change of 10^−8^ g/cm^2^. The authors compared the minimum detectable mass shifts by FBAR devices operated in liquid and gaseous environments against varying Q-values in air. They observed that a higher Q-value leads to improved sensitivity of FBAR device. Lin et al. [179] immobilized antibodies via protein A on the gold electrode of FBAR for highly sensitive detection of explosives. The designed sensor could be used without sample pre-concentration and showed high sensitivity for trinitrotoluene (TNT) and 1,3,5-trinitro-1,3,5-triazacyclohexane (RDX) using FBAR devices operated at 1.96 GHz and 1.65 GHz, respectively. High resonance frequency, use of specific receptors and miniaturized design leads to the development of smart sensors for detecting explosives that can be used for military and security applications. Benetti and coworkers [180] developed FBAR sensors for hydrogen (H_2_), carbon monoxide (CO) and ethanol vapors using two different chemical layers. For H_2_, they used Pd (layer thickness 15 nm) and for CO and ethanol, they selected co-tetra-phenyl-porphyrin (layer thickness 36 nm) following thermal evaporation process for their fabrication with 1.59 GHz FBAR device. The developed sensors showed the lowest detection concentrations of 2 ppm (H_2_), 40 ppm (CO) and 500 ppm (ethanol). Chen et al. [181] reported a dimethyl methyl phosphonate (DMMP) sensor integrating a self-assembled composite bilayer of Cu^2+^/11-mercaptoundecanoic acid with 2.35 GHz FBAR. The lowest measured concentration of DMPP was 100 ppb. Penza et al. [182] developed a nanocomposite of single walled carbon nanotubes (SWCNTs) embedded in cadmium arachidate by Langmuir–Blodgett method for gravimetric sensing of organic vapors. They optimized the composition of nanocomposite, i.e., the percentage of SWCNTs in cadmium arachidate for enhanced sensitivity toward ethyl acetate and toluene in their previous studies [183]. Table 4 shows the comparative sensor values of sensitivity and noise level for QCM, SAW and FBARs devices having 10 MHz, 433 MHz and 1045 MHz resonance frequencies, respectively. All the devices were coated with the same nanocomposite coating material having 75% SWCNTs. The data of QCM and SAW was compared from their previous work [183].

As expected, Table 4 shows that simply by increasing the resonance frequency of the acoustic device, the sensitivity increases for both ethyl acetate and toluene however, noise also increases. The authors proposed that with the increasing resonance frequency, there is a compromise between sensitivity and the noise. However, for a certain signal to noise ratio, the sensitivity increases with the increasing resonance frequency of acoustic devices. The observed detection limits were significantly low indicating the potential of designed sensor setup for room temperature sensing of volatile organic compounds (VOCs) at the workplace and others.

Hu et al. [184] developed polymer-coated FBAR sensor array for gas chromatographic analysis of different organic vapors. They integrated FBAR sensor array with a commercial gas separating column outlet and from other side connected to network analyzer equipped with a computer for data processing. The developed GC-FBAR sensor array combined with PCA demonstrated suitable selectivity pattern for quantitative analysis of different organic vapors. For instance, a dual mixture of acetone and n-pentane which is not easily separated by GC column, was effectively distinguished and quantified by PCA treatment of FBAR sensor array. In Table 5, the authors compared the performance of developed FBAR sensor array with different GC-sensor systems reported in the literature in terms of detection limit, dynamic range, linearity, application field and data processing algorithm. They suggested that high sensitivity of FBAR sensor array and its miniaturized design make it a potential candidate for microscale portable GC systems.

Guo et al. [190] presented a theoretical approach to study VOC sensor based on polymeric layer coated diaphragm integrated with FBAR. The polymer layer undergoes swelling on the absorption of analyte vapors that causes deformation of the diaphragm resulting in bending stress on piezoelectric film. Thus, a change in phase velocity and frequency of FBAR is observed. Simulation study revealed a sensitivity of 2.5 Hz/ppm and a detection limit of 0.4 ppm for chloroform vapors. Zhao et al. [191] reported a 1.5 GHz protein functionalized FBAR sensor for detecting a common insect repellent i.e., N,N-diethyl-*meta*-toluamide indicating the use of FBARs as odorant sensors. Song et al. [109] integrated polyethyleneimine modified SWCNTs with gold electrode surface via self-assembling for chemical sensing of formaldehyde vapors. The resonance frequency of FBAR device was about 4.5 GHz which resulted in an exceptionally low detection limit of 24 ppb and sensitivity of 1.472 kHz/ppb for formaldehyde. Furthermore, the developed sensor showed suitable selectivity when exposed to common interferents such as ethanol, acetone, benzene, toluene, chloroform and dichloromethane. The concentrations of these interferents were kept twice as high as compared to the corresponding formaldehyde concentrations. The result is shown in Figure 10 where it can be seen that developed sensor possesses high selectivity for formaldehyde sensing. The response and recovery times was less than 1 min including complete reversibility of sensor signal.

## 4. Advantages and Limitations

QCM is the one of the most extensively studied acoustic transducers for chemical/biochemical sensor applications because of its small size, easy availability, low temperature coefficient, low fabrication cost, and equally good sensitivity and suitability for gas as well as liquid phase operations. However, the usual operating frequency range is 5–20 MHz, which is much less than those of SAW and FBAR. The use of inverted mesa designs has led to the development of QCMs having resonance frequencies as high as 200 MHz with improved Q-values and energy trapping. Uttenthaler et al. [149] reported the first HFF-QCM biosensor for the detection of bacteriophage and since then some other research groups demonstrated their successful sensing applications in complex mixtures including both liquids and gaseous environments. Apart from these examples as discussed above, the number of research articles related to HFF-QCM sensors is relatively less. This somewhat indicates that further optimization in manufacturing is required to improve the oscillation stability issues in HFF-QCM and to solve the design and fabrication related concerns. And importantly, all this has to be managed in competitive price while compared to conventional acoustic transducers.

SAW devices are well known for gas phase sensing applications, their high fundamental resonance frequency, easy fabrication, compact design and ability for wireless integration are the main attractive features. SAW based sensor devices are one of the special classes of acoustic transducers that have gained the commercial success in gas phase analysis. The main limitation of typical SAW sensors is excessive damping problems in liquids which can be resolved by carefully selecting piezoelectric material for device fabrication and adjusting the cutting angle of the crystal. For example, SH-SAW or STW devices are frequently reported surface acoustic devices for sensor applications in liquids. Concerning operating frequency, the usual SAW resonators reported for chemical sensing have resonance frequency around 100–500 MHz which is much higher than HFF-QCMs. Although SAW devices with frequency as high as 10–14 GHz are reported [192] in the literature as pressure sensors, their chemical/biochemical sensing examples with such a high frequency are rare. SAW sensors with frequency around 1.5 GHz is reported for gas/vapor as well as liquid phase biosensor applications. An interesting strategy reported in literature is the use of higher harmonic modes to achieve higher frequency i.e., 6.4 GHz with enhanced Q-value of 4000. Despite the fact that a large number of articles related to fabrication of high frequency SAW devices i.e., in GHz range, their chemical/biochemical sensing applications are not as frequent as one would expect. The use of high frequency SAW devices combined with specific receptors, particularly for biosensing applications, is currently the focus of research to make these devices as successful as gas phase sensors.

In chemical/biochemical sensing, FBARs are newer acoustic devices than QCM and SAW. Their usual operating frequency is in the range of hundreds of MHz to 10 GHz which is substantially higher than QCM and reported SAW sensors. In sensor domain, there are two main advantages of using FBARs as acoustic transducer over QCM and SAW, the first is the high resonance frequency that leads to enhanced sensitivity and lower detection limits while the second is their ability to work in liquid mediums. Moreover, the much smaller size of FBARs i.e., around 200 µm in a typical cross-device size is advantageous for building a miniaturized sensor array that can be used for high throughput analysis. Their integration in microfluidic package can lead to develop lab-on-a-chip setup for clinical diagnostics and other applications. FBAR devices fabricated in an array combined with CMOS technology can be used for biosensor applications. In the recent examples as discussed above, FBAR devices demonstrated their suitability for gaseous as well as liquid phase sensing with many of them having resonance frequency above 1 GHz and the highest resonance frequency of 4.5 GHz. However, despite these advantages, the commercial manufacturing of FBAR for sensor applications is not as common as QCM and SAW devices which have a direct impact on its availability and price. Currently, FBAR fabrication is carried out by different techniques including sputtering, photolithography, electron beam lithography (EBL), and others. These are sophisticated nanofabrication tools that require special training/handling protocols for obtaining desired electrode geometry and dimensions which ultimately limit their mass production for using routine sensor applications.

## 5. Summary and Perspectives

In this review, we aimed to provide a concise overview of the high frequency acoustic transducers such as QCM, SAW and FBAR for a diverse range of chemical/biochemical sensor applications. We exclusively discussed their transduction mechanism, operating frequencies, piezoelectric materials used for fabrication, temperature-frequency dependency and operation in liquid environments. The exemplary chemical/biochemical sensor applications of exclusively high frequency QCM, SAW and FBAR devices are explained focusing on the two major domains i.e., biosensors for recognition of bio-relevant analytes in the liquid medium and other is the gas phase sensing. The orientation and integration of biological receptors as sensor coatings with acoustic transducers is also emphasized for improved recognition of target analyte in complex matrices. Finally, these acoustic transducers are compared in view of their advantages and limitations. From the above examples and comparisons, it is indicated that high resonance frequency has the pivotal role for achieving enhanced sensitivity and lower detection limits. However, with the increase in resonance frequency it is also important to maintain high Q-factor to reduce spurious signals and noise thus, to have improved oscillation stability. The comparisons between QCM, SAW and FBAR suggest that a tradeoff exists between the operating frequency of acoustic transducers and their availability and price. For instance, HFF-QCMs are getting more common for both gas and liquid sensing but has frequency around only 200 MHz. SAW resonators gained some commercial success as gas sensors but has certain limitations for the liquid phase operation, these devices with frequency around 1 GHz are reported for sensor applications. FBARs possess the highest fundamental frequency of > 4.5 GHz but are not manufactured as commonly as QCM and SAW. Current technological tools need further optimization and advancement in fabrication protocols to achieve high fundamental frequency with low manufacturing costs thus, to make these acoustic devices highly competitive transducers in the sensing world.

## Figures and Tables

**Figure 1 sensors-19-04395-f001:**
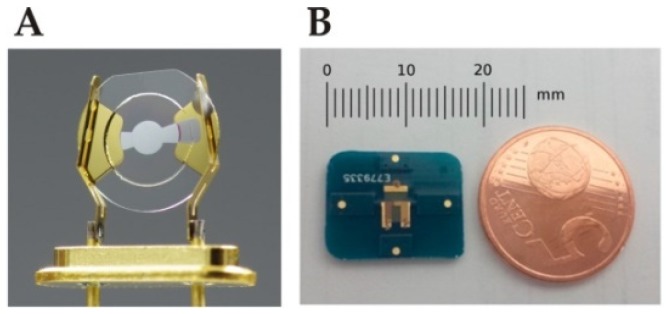
(**A**) Image of 195 MHz HFF-QCM from KVG Quartz Crystal Technology GmbH, (Neckarbischofsheim, Germany). The quartz crystal has diameter of approximately 5 mm, adapted from [93]. (**B**) A 150 MHz HFF-QCM manufactured by Advanced Wave Sensors (AWS) S. L. (Paterna, Spain). The device is integrated in a customized printed circuit board (PCB) to improve handling, PCB has dimension of 17.5 × 14 mm^2^ with thickness 1.55 mm, adapted from [94].

**Figure 2 sensors-19-04395-f002:**
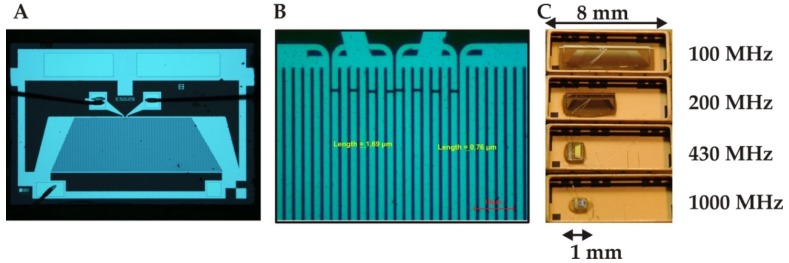
(**A**) A high resolution optical microscopic image of surface acoustic wave (SAW) resonator, adapted with permission from [102]. (**B**) The magnified image shows the geometrical details of IDTs i.e., the width of electrode is 1.69 µm and the distance between two consecutive fingers is 0.76 µm, adapted from [104]. (**C**) Images of different SAW resonators as going from 100 MHz to 1000 MHz, the size of devices changes from about 8 mm to 1 mm. This indicates that increase in fundamental resonance frequency of SAW demands more compact design.

**Figure 3 sensors-19-04395-f003:**
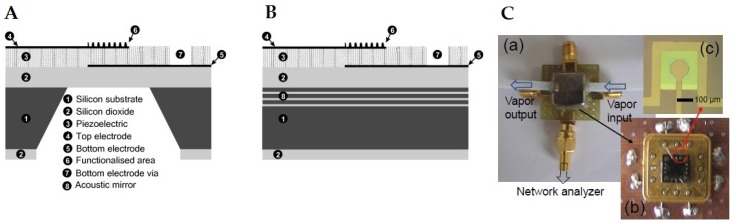
(**A**) Schematic design of a typical film bulk acoustic wave resonator (FBAR), (**B**) Schematic diagram of solidly mounted resonator (SMR), adapted with permission from [90]. (**C**) (**a**) Image of packaged FBAR sensor designed for sensing of formaldehyde, (**b**) FBAR device integrated with PCB, (**c**) magnified image of sensor element, adapted with permission from [109].

**Figure 4 sensors-19-04395-f004:**
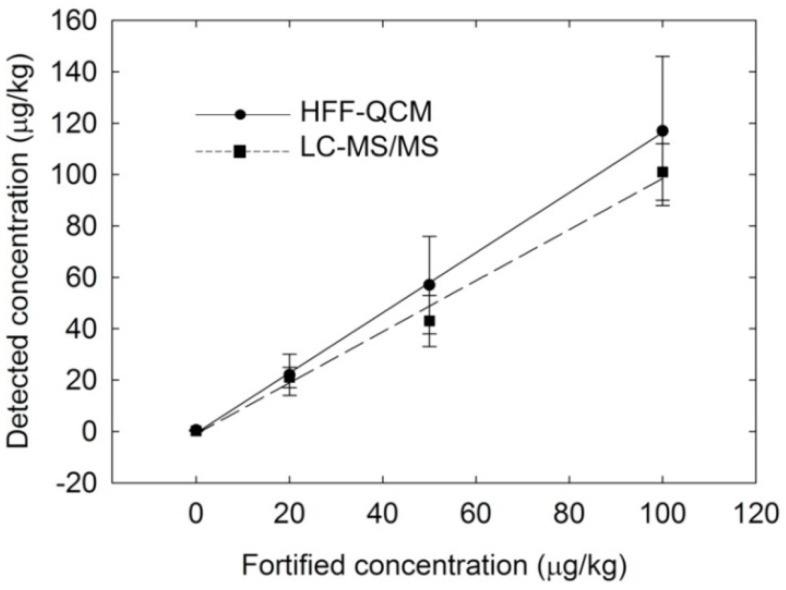
Analysis of the fortified honey samples by HFF-QCM sensor and LC-MS, the correlation coefficient for HFF-QCM was 0.999 and for LC-MS was 0.992. The linear regression slope for HFF-QCM was 1.14 and for LC-MS was 1.04, adapted with permission from [151].

**Figure 5 sensors-19-04395-f005:**
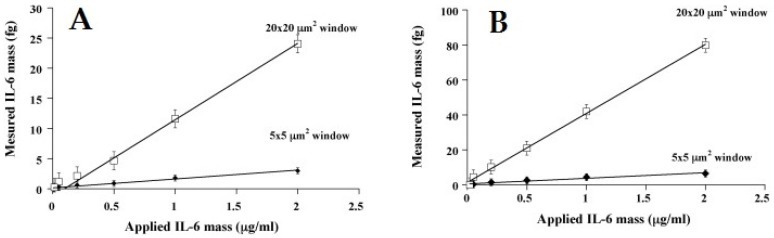
(**A**) SAW sensor response as measured IL-6 mass (fg/mL) vs. applied IL-6 mass (µg/mL), the sensor surface was modified following IL-6 immobilization through BSA. (**B**) SAW sensor response as measured IL-6 mass (fg/mL) vs. applied IL-6 mass (µg/mL), the sensor surface follows specific IL-6 binding through immobilization of IL-6 monoclonal antibodies. In both figures two windows of 5 × 5 µm^2^ and 20 × 20 µm^2^ were tested, sensor measurements indicated that by increasing the active surface area of the device, the sensor response increases as shown by 20 × 20 µm^2^ window in both cases, adapted with permission from [154].

**Figure 6 sensors-19-04395-f006:**
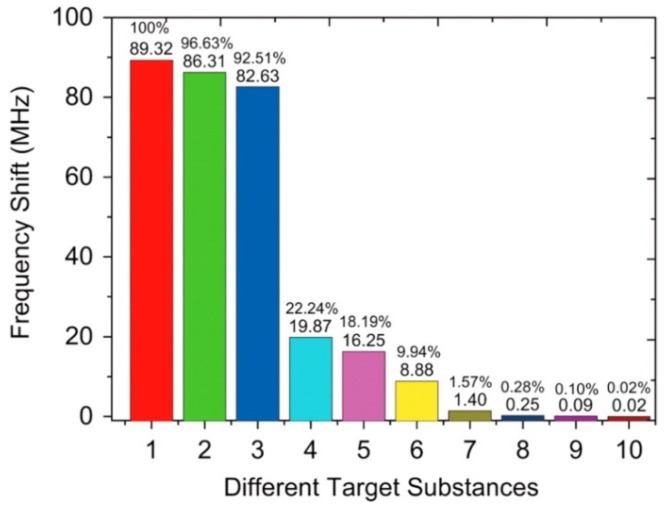
Comparison of SAW frequency shifts for different targets. Here, **1** is the response of target DNA (1 µg/mL) having 15 bases, **2** is the response of non-target DNA (1 µg/mL) having 15 bases with one mismatch base, **3** is the response of non-target DNA (1 µg/mL) having 15 bases with two mismatch bases, **4** is the response of non-target DNA (1 µg/mL) having only 4 bases, **5** is the response of non-target DNA (1 µg/mL) having only 3 bases, **6** is the response of target DNA (100 ng/mL) having 15 bases, **7** is the response of target DNA (10 ng/mL) having 15 bases, **8** is the response of target DNA (1 ng/mL) having 15 bases, **9** is the response for BSA, **10** is the response for PBS buffer, adapted with permission from [156].

**Figure 7 sensors-19-04395-f007:**
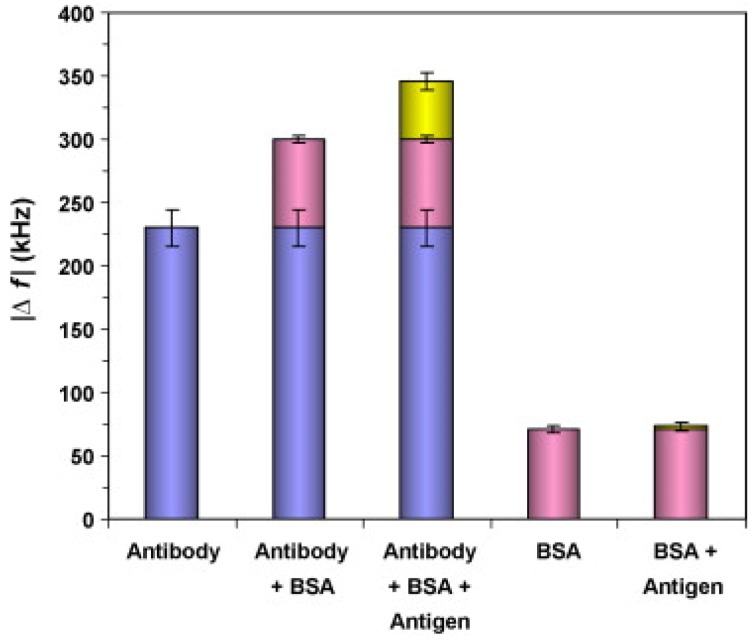
The frequency shift after h-PSA antibody adsorption (5 mg/L) for 15 min was 230 kHz (1st column), then BSA blocking (50 mg/L) for 25 min leads to 70 kHz shift (2nd column) and further antigen h-PSA binding (5 mg/L) for 15 min resulted another 46 kHz shift (3rd column). In control experiments, BSA adsorption at electrode surface (50 mg/L) for 30 min leads to 71 kHz shift (4th column) and then h-PSA antigen adsorption (5 mg/L) for 15 min over BSA resulted only 2 kHz shift (5th column), along *y*-axis Δf represents resonant frequency shifts, adapted with permission from [161].

**Figure 8 sensors-19-04395-f008:**
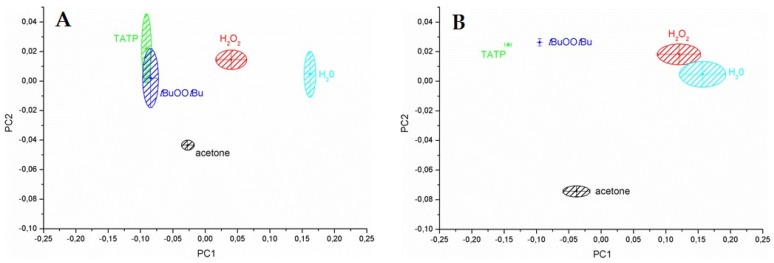
(**A**) PCA result of HFF-QCM sensor array for the detection of TATP without pretreatment, (**B**) PCA result of HFF-QCM sensor array for the detection of TATP after surface pretreatment, indicating that surface treatment resulted an improved TATP detection with better separation from interferents, adapted with permission from [166].

**Figure 9 sensors-19-04395-f009:**
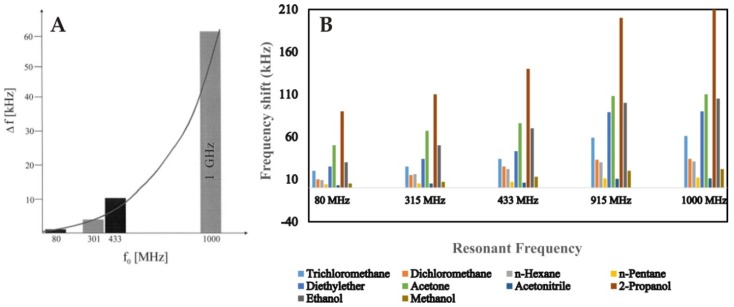
(**A**) Sensor responses of 80, 301, 433 and 1000 MHz SAW sensors against 1000 ppm of toluene vapors, along *x*-axis the f_o_ represents fundamental resonance frequency, all the devices were coated with permethylated β-cyclodextrin linked with hexafluorobenzene having thickness 60 nm, adapted with permission from [170]. (**B**) Sensor responses of 80, 315, 433, 915 and 1000 MHz SAW sensors against 100 ppm of different vapors, all the devices were coated with ZnO-CuO-CNT nanocomposite, adapted with permission from [171]. In both figures, sensor response increases as the resonance frequency of SAW resonator increases.

**Figure 10 sensors-19-04395-f010:**
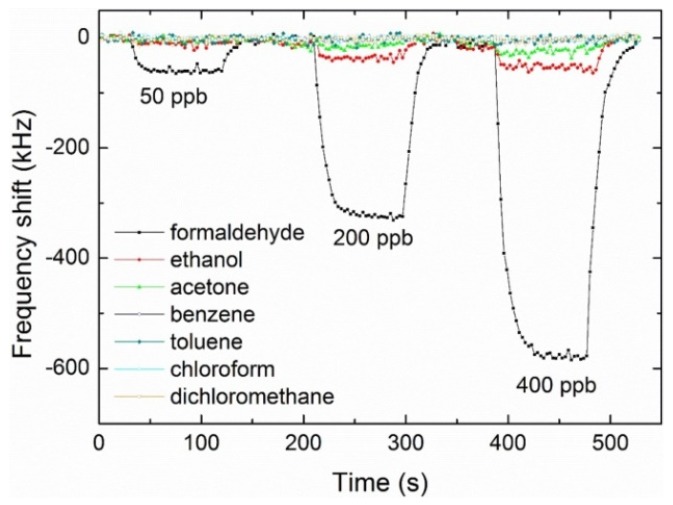
Sensor response of FBAR device coated with polyethyleneimine-modified single walled carbon nanotubes for formaldehyde vapors at 50 ppb, 200 ppb and 400 ppb, measured at 23 °C and 40% relative humidity. Sensor shifts to different interferents such as ethanol, acetone, benzene, toluene, chloroform and dichloromethane were also recorded at concentrations twice high as of formaldehyde, adapted with permission from [109].

**Table 1 sensors-19-04395-t001:** An overview of high frequency acoustic devices for chemical/biochemical sensor applications. The acoustic wave type, operating frequency, sensitivity, nature of piezoelectric materials, dielectric constants, acoustic wave velocity, temperature coefficient of frequency, working medium, availability, manufacturing costs, commercial success and general comments are described briefly.

	Quartz Crystal Microbalance (QCM)	Surface Acoustic Wave (SAW)	Film Bulk Acoustic Resonator (FBAR)
**Acoustic Wave Type**	Bulk	Surface	Bulk
**Operating Frequency**	5–200 MHz	100–1500 MHz	1.0–4.5 GHz
**Sensitivity**	Low	Intermediate	High
**Common Piezoelectric Materials**	Quartz	Quartz, LiNbO_3_, LiTaO_3_	AlN, ZnO
**Dielectric Constant**	Quartz = 3.8	LiNbO_3_ = 29 LiTaO_3_ = 43	AlN = 8.5 ZnO = 8.8
**Acoustic Wave Velocity (m/s)**	Quartz = 3158	LiNbO_3_ = 3488–3992 LiTaO_3_ = 3230–3394	AlN = 5600 ZnO = 2558
**Temperature Coefficient of Frequency (TCF) (ppm/°C)**	Quartz ≅ 0	LiNbO_3_ = 75–94 LiTaO_3_ = 35–64	AlN = 25 ZnO = 40
**Working Medium**	Liquids and Gases	Gases	Liquids and Gases
**Availability**	Common and Easily Available	Common and Easily Available	Not Easily available
**Manufacturing Costs**	Low	Moderate	High
**Commercial Adaptability**	Medium	High	-
**General Comments**	Most common and widely used acoustic transducer for sensor applications. QCMs have low fabrication costs. But has comparatively low sensitivity due to low resonance frequency.	Well known and widely used for gas phase sensor applications and found a number of commercial applications. However, not commonly used for liquid phase sensing.	Highly sensitive acoustic transducer for gaseous and liquid phase sensing. Not commonly available as advance fabrication techniques are need for their manufacturing thus, has high costs.

Acoustic wave velocity and TCF values for quartz, LiNbO_3_ and LiTaO_3_ are taken from [23]. The dielectric constant value of quartz is from [102] while for LiNbO_3_, LiTaO_3_, AlN and ZnO are from [147]. Acoustic wave velocity of AlN and ZnO are taken from [148] while TCF values are from [132].

**Table 2 sensors-19-04395-t002:** Comparison of analytical performance for carbaryl detection by ELISA, SPR and QCMs of variable frequencies, adapted with permission from [150].

Analytical Parameters (µg L^−1^)	ELISA		SPR	QCM: Fundamental Resonance Frequency (MHz)			
	Standard	Optimized		9	10	50	100
**LOD**	0.13	0.01	1.41	13.30	4.00	0.23	0.14
***I*_50_**	0.72	0.06	3.12	30.34	16.70	1.95	0.66
**WR**	0.23–2.36	0.02–0.18	1.91–5.75	18.30–50.30	7.00–35.00	0.50–7.20	0.26–1.72

**Table 3 sensors-19-04395-t003:** Comparison of developed SAW sensors for relative humidity sensing with other strategies reported in literature, adapted with permission from [173].

f_o_	Substrate	Surface Morphology	RH Range (%)	Max Response of Frequency Shift	Recovery & Response Time	Max Difference of Δf (kHz)	Ref.
1.54 GHz	128° YX LiNbO_3_	NiO nanoparticles	11–85	−5.81 MHz	4 s, 23 s	478	This work
1.54 GHz	128° YX LiNbO_3_	NSOH nanobelts	11–85	−2.95 MHz	10 s, 21 s	130	This work
395 MHz	ZnO/poly-imide (PI)	Graphene oxide	11–85	−1.5 MHz	5 s, 22 s	-	[174]
199 MHz	ST quartz	SiO_2_ films	30–93	−520 kHz	<10 s, <10 s	-	[175]
126 MHz	AlN/Si	Ga doped ZnO films	10–90	−400 kHz	-	20	[176]
433 MHz	ST quartz	Silicon-containing polyelectrolyte	10–95	−30 kHz	~10 s, ~10 s	-	[177]

**Table 4 sensors-19-04395-t004:** Comparison of operating frequencies, noise level and sensitivities of QCM, SAW and thin film bulk acoustic resonator (TFBAR) devices for chemical sensing of ethyl acetate and toluene vapors. All the devices were coated with 75 (wt.%) SWCNT-based nanocomposite Langmuir–Blodgett layers, adapted with permission from [182].

Acoustic Device	Operating Frequency (MHz)	Noise Level (Hz)	Sensitivity (ΔF/c) ΔF_w_ (Hz/ppm)/kHz	
			Ethyl Acetate	Toluene
**QCM**	10	1–5	0.05	0.097
**SAW**	433	10–25	0.30	1.020
**TFBAR**	1045	50,000	0.98	1.044

ΔF referred to frequency shift due to gas exposure; c is the concentration of gas; ΔF_w_ is the frequency shift due to coating onto device.

**Table 5 sensors-19-04395-t005:** Comparison of different GC-sensor systems with developed FBAR sensor in terms of limit of detection, dynamic range, linearity, application field and data process method, adapted with permission from [184].

GC-Sensor	Sensing Performance			Application Field	Data Process Method	Ref.
	Limit of Detection	Dynamic Range	Linearity			
Micro-cantilever	Sub-ppb	/	Good	Building-related illness	PCA	[185]
Semiconductor metal oxide detector	Sub-ppb	0.1–5 ppb	Non-linearity	Environmental pollution	/	[186]
Micro-capacitive detector	0.1 ppm	0.1–1 ppm	Good	Indoor pollutants	/	[187]
Acoustic wave (AW) device	/	/	Good	Food products and antiseptic agent	PCA	[188]
Nano-electromechanical resonators	Sub-ppb	0.6–1500 ppb	Good	/	/	[189]
Film bulk acoustic wave resonator	1 ppm	50–500 ppm	Good	Exhaled breath detection	PCA	This work

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
