# Peer review of "An Overview of High Frequency Acoustic Sensors—QCMs, SAWs and FBARs—Chemical and Biochemical Applications"

_sensors, 2019, doi:10.3390/s19204395_

Round 1

Reviewer 1 Report

According to authors, the paper provides “a concise overview of high frequency acoustic transducers such as quartz crystal microbalance (QCM), surface acoustic wave (SAW) and film bulk acoustic resonators (FBARs) to compare their working principles, resonance frequencies, selection of piezoelectric materials for their fabrication, temperature-frequency dependency and operation in the liquid phase.” The presentation is clear and well organized, with few exceptions. A compilation of 191 articles published in journals, proceedings and books, those entire specific to acoustic interrogation of gas/vapor and analyte properties for chemistry and biochemistry applications, is presented. The work should have some significance in the field of chemical and biochemical. The paper can be accepted for publication after taking into account the following minor concerns:

a) Authors should carefully read the manuscript again to find and correct all errors, and make a well-written article. Most of the time, I had to read a sentence twice to understand what the authors mean. Please ask a colleague who is fluent in English to edit your article before submitting the final version.

Correct use of commas and periods can make it clearer to understand some sentences in your text. For example (between rows 185 and 190): “The basic difference between these modes is the orientation of piezoelectric crystal, (COMMA) i.e. c-axis angle [107,108], (COMMA) as in the case of TLM WHERE (CONJUNCTION) the c-axis angle of crystal orientation is perpendicular to the substrate material. Whereas in TSM, the crystal is planar, (COMMA) i.e. inclined around 34.5_degrees, (COMMA) which produces shear waves. In TLM, the acoustic waves suffer from high attenuation loss when used in liquid media and thus, mainly used for gaseous phase sensing, (COMMA) while TSM devices exhibit low damping and thus, better suited for liquid phase operations.”

b) Idem, between rows 266 and 269: “As mentioned above, AlN offers high acoustic wave velocity that yields high frequency FBAR devices. (PERIOD) In addition to this, AlN interface is inert and biocompatible thus, suitable for integrating with biological receptor materials for biosensing applications.

c) The following sentence is confusing (between rows 221 and 223): “Such sub-micro size electrode geometries are somewhat difficult to realize by a conventional photolithographic process. (THIS PASSAGE IS HARD TO UNDERSTAND) Since the photolithographic methods are limited to electrode dimensions of around 0.5 micro_meter which yields SAW frequency of about 1 GHz.”

d) In the following items, you are asked to rewrite the sentences due to the difficulty of understanding them. In some sentences the comma/semicolon is missing, others lack conjunctions; there is misuse of the periods, etc. The sentence between rows 278 and 281 is confusing. Please rewrite it more clearly.

e) The sentence between rows 301 and 304 is confusing. Please rewrite it more clearly.

f) The sentence between rows 322 and 324 is confusing. Please rewrite it more clearly.

g) The sentence between rows 385 and 387 is confusing. Please rewrite it more clearly.

h) The sentence between rows 411 and 412 is confusing. Please rewrite it more clearly.

i) Please replace “Figure 8A and 8B demonstrate…” by “Figures 8A and 8B demonstrate…” in 576.

j) I think it is appropriate to separate the value of a quantity from your measuring units in: 195MHz (row 128), 3000m/s (row 221), and so on (throughout the manuscript);

k) As a suggestion, I would write the sentence (row 96) “According to Sauerbrey, fundamental…” as “According to Sauerbrey [26], fundamental…”.

l) Please write the name of the Scientist correctly on row 179 (Bragg).

m) I think there is a typo in the sentence (between rows 262 and 263): “Zinc oxide (ZnO) and [126] AlN [127] thin films…”

n) Please write the Scientist name with the first letter capitalized in row 353 (“In Love wave sensors…”)

o) I suggest using paragraph before presenting table 1 (row 372).

p) As you will use the acronym LOD in table 2, I suggest defining it on line 406: “…performance i.e. sensitivity, limit of detection (LOD) and working range of HFF-QCM immunosensor with…”

q) In the following sentence (row 429): “…In another effort [151], the authors combined…”, the reader is led to believe that this is a third paper written by the same group as [149-150], but it seems to me that this is not the case…

r) I suggest writing "… that the response of target DNA (1) having …" (row 471) as "… that the response of target DNA (Sample 1) having …", writing “1” in bold. Also, remove bold from “1” in “1 micro_g / mL” (row 472).

s) Please specify the variable “Delta_f” on the vertical axis of figure 7. Explain the need to use a module with this quantity.

t) Please specify “PC1” and “PC2” in figure 8. Please specify “VOC” acronym in row 600 (Volatile Organic Compounds?). Please specify the variable “f­­_zero” in horizontal axis of figure 9. Please specify the variables “Delta_F”, “Delta_Fw” and “c” in table 4.

u) There are some problems in References: when importing data about the articles used in paper, some acronyms or uppercase letters are incorrectly written in lowercase letters. See, for example, “saw” (reference 86), "hff-qcm" (reference 93), “LiTaO3” (sub-index in reference 103).

v) Reference [59]: the journal name, volume and pages are missing. These are easily found on Google.

w) According to your standard, in References section, all words in an article title begin with lowercase letters except the first one (exceptions, for names in English, acronym). This standard was not obeyed in references [133], [140], [167] and [186].

x) In the proceeding reference [117], word “In” is missing.

y) Error citing reference [177]: please correct “Hao, Z.; Eun Sok, K.” by “Zhang, H.; Kim, E.S.”

Author Response

According to authors, the paper provides “a concise overview of high frequency acoustic transducers such as quartz crystal microbalance (QCM), surface acoustic wave (SAW) and film bulk acoustic resonators (FBARs) to compare their working principles, resonance frequencies, selection of piezoelectric materials for their fabrication, temperature-frequency dependency and operation in the liquid phase.” The presentation is clear and well organized, with few exceptions. A compilation of 191 articles published in journals, proceedings and books, those entire specific to acoustic interrogation of gas/vapor and analyte properties for chemistry and biochemistry applications, is presented. The work should have some significance in the field of chemical and biochemical. The paper can be accepted for publication after taking into account the following minor concerns:

We are thankful to reviewer for his comments. The point by point responses to reviewer comments are described below.

a) Authors should carefully read the manuscript again to find and correct all errors, and make a well-written article. Most of the time, I had to read a sentence twice to understand what the authors mean. Please ask a colleague who is fluent in English to edit your article before submitting the final version.

Correct use of commas and periods can make it clearer to understand some sentences in your text. For example (between rows 185 and 190): “The basic difference between these modes is the orientation of piezoelectric crystal, (COMMA) i.e. c-axis angle [107,108], (COMMA) as in the case of TLM WHERE (CONJUNCTION) the c-axis angle of crystal orientation is perpendicular to the substrate material. Whereas in TSM, the crystal is planar, (COMMA) i.e. inclined around 34.5_degrees, (COMMA) which produces shear waves. In TLM, the acoustic waves suffer from high attenuation loss when used in liquid media and thus, mainly used for gaseous phase sensing, (COMMA) while TSM devices exhibit low damping and thus, better suited for liquid phase operations.”

Response:

We are thankful to reviewer for critically reading the article and pointing the grammatical and formatting mistakes. These sentences are rephrased as suggested by reviewer. Furthermore, we have also revised manuscript thoroughly.

b) Idem, between rows 266 and 269: “As mentioned above, AlN offers high acoustic wave velocity that yields high frequency FBAR devices. (PERIOD) In addition to this, AlN interface is inert and biocompatible thus, suitable for integrating with biological receptor materials for biosensing applications.

Response:

The sentence is rephrased as suggested by reviewer.

c) The following sentence is confusing (between rows 221 and 223): “Such sub-micro size electrode geometries are somewhat difficult to realize by a conventional photolithographic process. (THIS PASSAGE IS HARD TO UNDERSTAND) Since the photolithographic methods are limited to electrode dimensions of around 0.5 micro_meter which yields SAW frequency of about 1 GHz.”

Response:

Thanks for reviewer’s comment; this sentence is rephrased for better understanding.

d) In the following items, you are asked to rewrite the sentences due to the difficulty of understanding them. In some sentences the comma/semicolon is missing, others lack conjunctions; there is misuse of the periods, etc. The sentence between rows 278 and 281 is confusing. Please rewrite it more clearly.

Response:

These sentences are rephrased for better understanding.

e) The sentence between rows 301 and 304 is confusing. Please rewrite it more clearly.

Response:

The sentence is rephrased for better understanding.

f) The sentence between rows 322 and 324 is confusing. Please rewrite it more clearly.

Response:

The sentence is rephrased for better understanding.

g) The sentence between rows 385 and 387 is confusing. Please rewrite it more clearly.

Response:

The sentence is rephrased for better understanding.

h) The sentence between rows 411 and 412 is confusing. Please rewrite it more clearly.

Response:

The sentence is rephrased for better understanding.

i) Please replace “Figure 8A and 8B demonstrate…” by “Figures 8A and 8B demonstrate…” in 576.

Response:

This has been corrected as Figures 8A and 8B.

j) I think it is appropriate to separate the value of a quantity from your measuring units in: 195MHz (row 128), 3000m/s (row 221), and so on (throughout the manuscript);

Response:

We agree to reviewer, we have separated the quantities form measuring units at row 128 and 221.

k) As a suggestion, I would write the sentence (row 96) “According to Sauerbrey, fundamental…” as “According to Sauerbrey [26], fundamental…”.

Response:

In revised version, we have written as “According to Sauerbrey [26], fundamental…”.

l) Please write the name of the Scientist correctly on row 179 (Bragg).

Response:

In revised version, we corrected it as Bragg.

m) I think there is a typo in the sentence (between rows 262 and 263): “Zinc oxide (ZnO) and [126] AlN [127] thin films…”

Response:

In revised version, we have corrected it as “Zinc oxide (ZnO) [126] and AlN [127] thin films…”

n) Please write the Scientist name with the first letter capitalized in row 353 (“In Love wave sensors…”)

Response:

In revised version, we have written as “In Love wave sensors”…..

o) I suggest using paragraph before presenting table 1 (row 372).

Response:

In revised version, we used paragraph before presenting table 1.

p) As you will use the acronym LOD in table 2, I suggest defining it on line 406: “…performance i.e. sensitivity, limit of detection (LOD) and working range of HFF-QCM immunosensor with…”

Response:

In revised version, the limit of detection (LOD) is defined near line 406.

q) In the following sentence (row 429): “…In another effort [151], the authors combined…”, the reader is led to believe that this is a third paper written by the same group as [149-150], but it seems to me that this is not the case…

Response:

We have gone through the references [149,150] and [151]. All three papers have at least two common co-authors and furthermore, all three papers have common affiliation from Universitat Politècnica de València. Therefore, they could be considered as the same group.

r) I suggest writing "… that the response of target DNA (1) having …" (row 471) as "… that the response of target DNA (Sample 1) having …", writing “1” in bold. Also, remove bold from “1” in “1 micro_g / mL” (row 472).

Response:

In revised version, we have changes it as “….that the response of target DNA (sample 1) having….”. Furthermore, the bold “1” in “1 µg/mL” is also removed.

s) Please specify the variable “Delta_f” on the vertical axis of figure 7. Explain the need to use a module with this quantity.

Response:

In figure 7, the variable “ΔF” along y-axis represents resonant frequency shift and as the module with this quantity was used in original figure and here we only reproduced the figure 7 as such without any change.

t) Please specify “PC1” and “PC2” in figure 8. Please specify “VOC” acronym in row 600 (Volatile Organic Compounds?). Please specify the variable “f­­_zero” in horizontal axis of figure 9. Please specify the variables “Delta_F”, “Delta_Fw” and “c” in table 4.

Response:

The terms PC1 and PC2 are generally referred to first principal component and second principal component, respectively which are used to explain the variability of the sensor data. The terms PC1 and PC2 are described in revised version of manuscript and we have also specified VOC as volatile organic compounds. Furthermore, the fo in x-axis of figure 9 is specified as fundamental resonance frequency. In table 4, we have specified the terms ΔF, ΔFw and c. The term “ΔF” is the frequency shift upon gas exposure, “ΔFw” is the frequency shift due to coating onto device and “c” is the concentration of gas.

u) There are some problems in References: when importing data about the articles used in paper, some acronyms or uppercase letters are incorrectly written in lowercase letters. See, for example, “saw” (reference 86), "hff-qcm" (reference 93), “LiTaO3” (sub-index in reference 103).

Response:

In the revised manuscript; we corrected it as SAW in reference [86], HFF-QCM in reference [93] and LiTaO3 in reference [103].

v) Reference [59]: the journal name, volume and pages are missing. These are easily found on Google.

Response:

In the revised manuscript; the journal name, volume and page numbers of reference [59] are written.

w) According to your standard, in References section, all words in an article title begin with lowercase letters except the first one (exceptions, for names in English, acronym). This standard was not obeyed in references [133], [140], [167] and [186].

Response:

In the revised manuscript; we have corrected the references [133], [140], [167] and [186].

x) In the proceeding reference [117], word “In” is missing.

Response:

In the revised manuscript; we have written “In” and corrected the reference [117].

y) Error citing reference [177]: please correct “Hao, Z.; Eun Sok, K.” by “Zhang, H.; Kim, E.S.”

Response:

In the revised manuscript; we have corrected reference [177].

Reviewer 2 Report

In this work, the authors  overview the development in  high frequency acoustic sensors  with 184 refs. This work is well compared and can be considered to publish after a minor revision. My  comments is listed below.

1)For QCM sensor,  the  a new reported associated resonant model has obvious  impact on its response, especially in high frequency region. (Unfound Associated Resonant Model and Its Impact on Response of a Quartz Crystal Microbalance in the Liquid Phase ,  Chem. 2018, 90, 2796-2804.). Some comments is needed to add in the text.

2)The influence of longitudinal wave on the response of QCM in liquid phase is not concerned, which is important to obtain reliable information.

Author Response

In this work, the authors overview the development in high frequency acoustic sensors with 184 refs. This work is well compared and can be considered to publish after a minor revision. My comments is listed below.

We are thankful to reviewer’s remarks and suggestions. The responses to reviewer’s comments are described below.

1)For QCM sensor, the a new reported associated resonant model has obvious impact on its response, especially in high frequency region. (Unfound Associated Resonant Model and Its Impact on Response of a Quartz Crystal Microbalance in the Liquid Phase, Chem. 2018, 90, 2796-2804.). Some comments is needed to add in the text.

Response:

Thanks for reviewer’s comments, in the revised manuscript; we explained the associated high frequency resonance model for QCM operation in liquid phase and cited this important article.  

2)The influence of longitudinal wave on the response of QCM in liquid phase is not concerned, which is important to obtain reliable information.

Indeed, we agree to the reviewer’s comments that the longitudinal wave effect is important for QCM oscillation in the liquid phase as the longitudinal waves are related to change in density and viscosity of liquid.

Reviewer 3 Report

Since this is a review article, the captions in the images should be written as following

"..adapted with permission from [ref]". 

GC detectors/sensors listed in Table 5 does not include some important types of detectors (e.g micro-plasma and chemi-resistors). I would recommend that authors include these types of detectors to provide complete comparison between different detection technologies.  

Author Response

Since this is a review article, the captions in the images should be written as following

"..adapted with permission from [ref]".

Response:

In the revised version, the captions of the images and tables are written as “adapted with permission from [ref].

GC detectors/sensors listed in Table 5 does not include some important types of detectors (e.g micro-plasma and chemi-resistors). I would recommend that authors include these types of detectors to provide complete comparison between different detection technologies.

Response:

We understand reviewer’s concern on adding some other types of GC detectors for complete comparison between different detection technologies however; the table 5 is reproduced as such from J. Hu et al. Sensors & Actuators: B. Chemical 274 (2018) 419–426 therefore, we cannot make any modifications in this table.